# Study on the Overall Reaction Pathways and Structural Transformations during Decomposition of Coal Fly Ash in the Process of Alkali-Calcination

**DOI:** 10.3390/ma14051163

**Published:** 2021-03-02

**Authors:** Jingjie Yang, Hongjuan Sun, Tongjiang Peng, Li Zeng, Li Chao

**Affiliations:** 1Key Laboratory of Ministry of Education for Solid Waste Treatment and Resource Recycle, Southwest University of Science and Technology, Mianyang 621010, China; pengtongjiang@swust.edu.cn (T.P.); zengli6041@mails.swust.edu.cn (L.Z.); chaoli001@mails.swust.edu.cn (L.C.); yangjingjie007@mails.swust.edu.cn (J.Y.); 2Institute of Mineral Materials and Application, Southwest University of Science and Technology, Mianyang 621010, China

**Keywords:** coal fly ash, alkaline calcination, FTIR, functional groups

## Abstract

In this research, phase transformation and the role of NaOH on the structure of coal fly ash (CFA) during an alkali-calcination process were identified by a combination of X-ray powder diffraction (XRD), Fourier transform infrared (FTIR) and deconvolution analysis. The variation in the different functional groups and structural parameters of the raw and post-alkali calcinated CFA were analysed by deconvolution of the FTIR results, conducted with a Gaussian approach. The results, firstly, provide a deep insight into the functional groups in CFA. In CFA systems, the vibration signals of Q^0^, Q^1^, Q^2^ and Q^3^ were detected and the dominant structural units associated with Si tetrahedron groups were isolated to Q^3^ and Q^2^. Deconvolution analysis of the band from 400 to 1400 cm^−1^ showed that the added NaOH resulted in an increase in Q^1^ at the cost of Q^3^ and Q^2^ and the degree of reaction of the CFA was, therefore, decreased. Concurrently, it was established that the changes in the Gaussian peak component were related to the calcination temperature and time that allowed us to tailor the model of the structural decomposition of CFA.

## 1. Introduction

Coal fly ash (CFA) is a typical coal-based solid waste that is produced in enormous amounts [1,2]. The production of CFA is expected to increase with economic increases, particularly in China, where the economy has displayed spectacular gross domestic product (GDP) growth in the last 30 years [3]. Currently, the inadequate processing of CFA can cause serious harm to cultivated land, groundwater pollution and regional air pollution of CFA landfills [4,5,6]. Hence, production, storage and transportation of CFA have been rigorously enforced by environmental protection regulations in order to reduce the damages as much as possible. Recycling of secondary resources has become essential with the depletion of primary resources [7]. The majority of CFA is used for constructing roads, the production of building sectors (bricks, blocks, tiles, etc.) [8] and in the concrete and cement manufacturing industries [9]. The increasing enforcement of environmental standards has forced researchers around the world to earnestly explore the development of high-quality treatments for CFA. Nevertheless, the progress of research for treating CFA is far behind that concerning the application of CFA in construction because of the complicated physio-chemical properties of CFA [10].

CFA is an incredibly complex anthropogenic material to characterize [11]. The characteristics of CFA are inherently heterogeneous, mainly depending on upstream raw materials, fuel combustion equipment, gas purification equipment and quenching process [12,13]. The CFA structure corresponds to a micro-crystal model, which means that it contains some small-scale micro-crystals in the matrix of its glass phase. It is also broadly described as a mixture of poor crystalline phases with hematite, magnetite, mullite quartz, as well as aluminosilicate glass phases [14]. The major constituents (Si, Ca, Al, K, Fe and so on) and trace elements (As, Ba, Co, Ni, U, Zr and other heavy metals) [15] occur as minerals and compounds in glass and other minerals and phases [16]. Another aspect of the inherent heterogeneities is that CFA often has a surface layer thickness ranging from nm to μm, formed from the readily-leachable material deposited while cooling [17]. In fact, the chemical composition and structure can be different between the outer layer and the interior of a single particle depending on the thermal history of the CFA [18]. The structure of the amorphous, or glassy, metal is a crucial parameter that defines the reactivity of the CFA [19]. It is known that the main glass network formers are Si^4+^ and Al^3+^ [20], and that these cations act as network modifiers in the CFA glass network [21]. Alkali calcination technology is an advanced and effective pre-treatment technique, during which the strong bonds between the silica and the metal within the compound can be disrupted. This allows a more effective release of the metal atoms from the SiO_4_ tetrahedrons [22]. Therefore, some studies [23,24,25] of hydrometallurgical alkaline explored approaches to removal silica from CFA. It is worth mentioning that there were indisputable benefits for the alkali-calcination step form shortening the time for zeolitisation and reducing the inclusion of amorphous-crystalline textures, thereby helping to purify the zeolite crystal phase [26,27,28]. Geopolymers are generally synthesized by conventional alkali mixing methods [29]. Additionally, researchers have obtained geopolymers after geopolymerisation of alkali calcinated CFA at a rate of 83.941 m^2^/g [30]. Alkali calcination reduces the complexity of the binder phases and reduces the amount of unreacted raw material, resulting in faster geopolymerisation. Furthermore, alkali calcination plays an important role in the high-efficiency recovery of valuable metals, especially REE from CFA [31]. It was noted that the leaching rate after alkali calcination was higher than that of conventional direct leaching [32]. Despite this progress, no in-depth research has been conducted to clarify the detailed chemical structure of CFA during the alkali calcination and there is still a lack of knowledge about the relative effects of experimental conditions on the evolution of the microstructure. In this study, fourier transform infrared spectroscopy (FTIR) was used to detect the slight changes occur in the bonding environment surrounding functional groups, and the variation in the functional groups was semi-quantitatively analysed by deconvolution of the spectrum. Furthermore, X-ray powder diffraction (XRD) and scanning electron microscopy (SEM) were used to determine phase transformation and changes of microstructure of CFA in the process of calcination.

## 2. Experimental

### 2.1. Sample Materials and Their Characterization

CFA Samples used in this research were collected from Da Tang coal-fired power plant located in Inner Mongolia of China. Analysis with an X-ray fluorescence spectrometer showed that the CFA was mainly composed of Al_2_O_3_ and SiO_2_, which together accounted for over 85.94% of the CFA. The chemical composition of the CFA was shown in Table 1. Due to the low Ca content, the CFA was classified as low calcium siliceous CFA. Table 1 also shows the Loss on ignition (LOI) of CFA was 3.41%, a low LOI value indicates that the CFA contains a large proportion of mineral ash.

The size distribution of the raw CFA was measured by a laser particle size analyser (Zetasizer Nano Zs90, Malvern, UK). Figure 1a shows that particle of sizes less than 6.663, 21.13, 58.85, 122.8 and 193.2 μm were present in 10%, 25%, 50%, 75% and 90% of the CFA, respectively.

It can be seen from Figure 1b that mullite was the main mineral component of the CFA. In addition, some peaks that appeared in the XRD spectrum, matched well with the standard quartz (SiO_2_) structure. XRD patterns also show the presence of hematite. A broad diffraction “hump” was observed in the region of 15° to 30° indicating the presence of an amorphous glass phase in the raw CFA [33].

Analytically pure NaOH was used to roast the CFA at high temperatures. All reagents are made by Kelong Chemical Co. Ltd., Chengdu, China. Deionised water was used in all the experiments.

### 2.2. Experimental Procedure and Apparatus

The raw CFA was oven-dried to a constant weight at 105 °C for 24 h. In each experiment, 2 mL deionised water was added to increase the plasticity of the mixture, which was prepared by thoroughly mixing the CFA and NaOH using a mortar and pestle. Then, the samples were pressed in cylindrical steel dies with 30 mm diameter and 10 mm thickness, using a pressure of 8 MPa. This was done in order to prevent sintering behaviour between the sintered body and the corundum crucible at high temperatures. Next, when the temperature of the muffle furnace reached a predetermined temperature, the samples were placed in a cylindrical alumina crucible and isothermally calcined with free access to air. The reaction process was stopped after different lengths of time which points the molten product was rapidly removed from the muffle furnace and cooled in a dryer. Table 2 shows the mix designs adopted for the mixture along with their corresponding CFA/NaOH mass ratios and the sample labels are suffixed by the corresponding calcination temperature and calcination time. Finally, all of the cooled alkali-calcination products were ground and leached in water for 120 min at 60 °C, with continuous stirring. At the end of each leaching experiment, the slurry was filtered via a vacuum filter. The insoluble residues were washed with deionised water until a neutral pH was reached, then the remaining residues were dried and sampled for analysis.

### 2.3. Characterization Techniques

The X-ray diffractometer (XRD, Ultima IV, Akishima, Japan) was utilized to analysis the difference of samples after alkali-calcination and the phases of CFA. The surface functional groups of CFA and its residues being leached with deionized water were analyzed by Fourier transform infrared spectroscopy (FTIR, Nicolet-5700, Madison, WI, USA) from 400 to 4000 cm^−1^ by means of the KBr tablet method. The microstructural development and chemical characteristics of samples were studied by a scanning electron microscope (SEM, Zeiss Instruments, Oberkochen, Germany) equipped with energy dispersive spectroscopy (EDS, Oxford, UK). To obtain clearer SEM pictures, the surface of samples was treated by gold plating, an EDS microanalysis instrument operating at an accelerating voltage of 10 kv in a high vacuum mode.

## 3. Results and Discussion

### 3.1. Analysis for Activated CFA by NaOH Calcination

In Figure 2, XRD results of the starting-CFA, reference samples calcined at 400 °C and 600 °C were presented, the mass ratio of the NaOH to CFA was 1:4. The mullite phase almost completely disappeared when the calcination temperature at 400 °C. Moreover, the quartz almost disappeared, while sodium aluminosilicate appeared in samples. This indicates that the mullite starts transforming in to the active NaAlSiO_4_ phase when the temperature at 400 °C. Additionally, as shown in Figure 2a,b, the Na_2_CO_3_ phase were detected. A possible explanation is that as the calcination time increased, further destruction of the CFA aggregation occured, which was achieved by increasing the exposure of unburned carbon that could react with NaOH. Furthermore, the excess NaOH might have reacted with CO_2_ in the air to form Na_2_CO_3_.

The XRD diffractions patterns of the water-leached residues were compared between Figure 2c,d. The mentioned hump in the diffraction patterns moved towards the 2θ angle positions approximately between 20° and 35°, while the intensity decreased. This change is related to the formation of a reaction compound in the amorphous phase [34,35]. The strong peaks belonging to quartz and mullite present in the raw CFA became significantly weaker in the diffractions of water-leached residues, which further indicates that quartz and mullite can be transformed into water-soluble phases at 400 °C, and the extraction rates were 8.62% and 9.86% for alumina and silica, respectively. The chemical composition of water-leached residues was shown in Table 3. There were also some differences in the XRD diffractions with the increase in temperature. The phase changes shown in Figure 2d indicate that nepheline appeared with the increase in the calcination temperature. It also presented broad peaks at 20°–35°. The phase changes shown in Figure 2d indicate that nepheline appeared with excessive NaOH addition under the higher calcination temperature (600 °C). Therefore, it can be concluded that sodium aluminosilicate is preferentially formed. Based on the above analysis, the following reactions were predicted to occur during the calcination experiment.

Reaction temperature: 400–600 °C.
Al_6_Si_2_O_13_(mullite) + 6NaOH + 4SiO_2_ → 6NaAlSiO_4_(nepheline) + 3H_2_O(1)
Al_6_Si_2_O_13_(mullite) + 6NaOH + 10SiO_2_ → 6NaAlSi_2_O_6_ + 3H_2_O(2)

### 3.2. FTIR Characterization

The functional groups of raw CFA, S4-1, S4-2, S4-3, S4-4, S6-1, S6-2, S6-3 and S6-4 can be determined from the FTIR spectra in Figure 3. As shown in the Figure 3a. The IR region at 3446 cm^−1^ consists of vibrations due to water, hydroxyl or similar groups [36]. The peak at 1644 cm^−1^ was assigned to molecular water [37].

There were characteristic peaks in the range 800–1200 cm^−1^, which corresponded to the Si-O stretching vibration in the Q^n^ units, where n refers to the bridging oxygen, known as Q^4^, Q^3^, Q^2^, Q^1^ and Q^0^ [38,39]. The existence of broad absorption bands means the long-range disorder in CFA structure and the wide existence of Q^n^ units [40]. On the other hand, lower wavenumbers (1000 cm^−1^) could also have been caused by the incorporation of a larger amount of Al into the silicate backbone [41,42]. The FTIR spectra showed a decrease in absorption bands at 719 cm^−1^, which was attributed to the asymmetric and symmetric vibrations of Si-O_b_-Al (O_b_ stands for bridging oxygen) [43]. These changes are consistent with more Si-O_b_-Al groups being broken as the temperature increased, and with the presence of more Q^n^ units. The FTIR spectra clearly indicated that decomposition of the CFA silicate network occurred. XRD analysis showed that sodium aluminiosilicate (a type of Q^0^ silicate species) appears when the Si-O bonds are decomposed by NaOH. Moreover, a series of signals in the 611 cm^−1^, typical of an Al-O band from aluminium-oxygen octahedron (AlO_6_) stretching vibrations [44].

The FTIR absorption spectra of S6-1, 2, 3 and 4 are shown in Figure 3b. In the spectra of S6-1 and S6-2, the most intense absorption bands lay in the 750–1250 cm^−1^ region, the next most intense between 540 and 750 cm^−1^ and the least intense between 1350 and 1550 cm^−1^. Note that the FTIR analyses of S6-3 and S6-4 showed a decrease in absorption bands in the 750–1250 cm^−1^ region as the calcination time increased. In addition, the FTIR analyses of S6-4 and S6-3 showed more consistent vibrations appearing in the range of 1423 cm^−1^, where the peaks appeared sharper than the vibrational peaks of S6-1 and S6-2. This band is related to the formation of nepheline by the reaction of NaOH with CFA. A detailed deconvolution analysis of the FTIR spectra and reasonable interpretation of the resolved bands can further help in understanding the various structural units of CFA.

### 3.3. Deconvolution Methods

In principle, the measured FTIR spectra represent a combination of multiple overlapping peaks. The relative amount of particular types of atomic bonds and structures can be estimated from the relative area of these peaks [45]. The main band corresponding to the Si(Al)-O-Si bond asymmetric stretching vibrations was more thoroughly investigated to acquire a more comprehensive understanding of the effect of NaOH on raw CFA during the alkali calcination process. An empirical deconvolution method was carried out, and the overlapping spectra are studied in more detail. This involved fitting the overlapping peaks generated in the 800 to 1300 cm^−1^ spectral range to a theoretical Gaussian curve. The R^2^-value of the combination of the deconvolved peaks to the original peaks was greater than 0.99.

The Gaussian bands fitted to the IR spectra of CFA, S4-1, S4-2, S4-3 and S4-4 are shown in Figure 4. It could be seen from the deconvolution results that the 800–1300 cm^−1^ band was characterized by multiple sharp peaks, indicating the diversity of functional groups. The absorption signal that occurred between 1139 and 1161 cm^−1^ was associated with asymmetric stretching of Si (Al)-O-Si in mullite or mullite-like structures [46]. The component bands located around 1100 cm^−1^ were associated with the presence of Q^3^ units in the amorphous glasses [47]. Respectively, as the activation time increased, both peaks decreased in intensity. This could be ascribed to the increase in metal cations in the outer body of the glass network breaking the Si-O and Al-O. The absorbance ratio of the symmetric stretching of Si-O-Si in the Q^3^ unit and asymmetric stretching of (Si, Al)-O-Si in the amorphous glasses were also observed to decrease with increased activation time, which reveals that the number of functional groups decreased after calcination. The peak assignments of the deconvolved bands in the 800 to 1300 cm^−1^ range of the spectrum are given in Table 4.

Comprehensive consideration of the number of functional groups consumed in the 800–1300 cm^−1^ band reflected the degree of three-dimensional structural degradation in the samples. To quantify the active bonds, it was assumed that the relative areas of the resolved bonds were proportional to their concentration in the samples. Overall, the content of functional groups after the alkali calcination was significantly reduced. According to Figure 5 and the FTIR spectra deconvolution data, it can be seen that (Si, Al)-O-Si in the amorphous glasses, followed by Si-O-Si in the Q^2^ and Q^3^ units, were the most consumed functional groups during the alkali calcination process. The (Si, Al)-O-Si in the Q^3^ units were largely consumed. Alkali calcination removes various functional groups from the CFA by breaking the chemical bonds. Therefore, Q^3^ structures were the most active functional groups in the alkali calcination process, followed by Q^2^ structures. The obtained results suggest that the glass partially denatured with NaOH, losing a large fraction of its chain-like and branched-chain structures. Consistent with previously reported studies [49], the network of the CFA particles broken under the action of OH^−^. Moreover, as the concentration of OH^−^ increases, the damage to the network became more intense [50]. Therefore, this clearly indicates that destruction of the aluminosilicate network occurred in the glass.

Standard procedures for resolving the FTIR curve of the samples were followed to obtain the deconvoluted spectra to gain detailed information on the functional groups and their linkages in the studied samples (S6-1, S6-2, S6-3 and S6-4). Figure 6 shows the deconvoluted spectra and the assignment of bands in the 1300–800 cm^−1^ region. Based on the fitted curve, four bands were distinguished. Hence, the deconvolution of the spectra provides information about the complex composition of different bands that had been masked in the unresolved spectra [51]. These masked bands occurred at about 1110, 1020, 959 and 849 cm^−1^. It was also observed that there was good correspondence between the above absorption peak positions of the deconvoluted spectra and those in the original spectra of the studied samples. The absorption peak intensity of S6-1 and S6-2 was significantly higher than that of S6-3 and S6-4, which reveals that the number of functional groups has a certain impact on the calcination additives at higher temperatures.

A peak was observed in S6-1 centred at 1110 cm^−1^ (Figure 6a), which is due to asymmetric stretching of the Si-O-Si in Q^3^ units. This peak was also present in S6-2, S6-3 and S6-4. The area of this peaks was higher in S6-1 and S6-2 compared to S6-3 and S6-4 (Figure 7). When the glass structure was broken in the alkali calcination reactions, the peaks related to Q^3^ units reduce in the CFA. The peak at 1020 cm^−1^ which was attributed to asymmetric stretching of (Si, Al)-O-Si in the amorphous glasses reduced as the reaction time increased, The variation in the wavenumbers of the samples shows the differences in the Al substitution in the structure of the studied samples. Symmetric stretching of Si-O-Si in the Q^2^ units was the main contribution to the 955 cm^−1^ area bands. The peak areas were found to vary in an identical manner as already discussed earlier in the results. The availability of Si and Al in the alkali calcination reactions could be the reasons for reduced functional groups.

### 3.4. SEM and EDS Analysis

SEM analysis was conducted on the ultrapure washed samples (Figure 8). A zoomed image taken at 2500× magnification showed large-diameter spheres enclosing sub-microspheres. The SEM images also showed some irregularly shaped particles, cohesive bodies, and debris of spheres; these materials appeared more irregular. The surfaces of the CFA spheres were relatively smooth at the scale of observation allowed by the SEM, regardless of the particle size. Morphological changes were observed with the change in calcination time. With increasing reaction time, the number of spherical particles started to decrease. Although the larger particles kept their ball-like skeletal structure, they exhibited cracks on their surfaces at high magnification (Figure 8b). This also indicated that there was an abundant amount of NaOH dissolved Si and Al on the surface of the CFA under high-temperature conditions. Under the elevated temperature, NaOH tended to melt and form liquid phases, and the increase in fluidity resulted in the further destruction of the CFA structure. It can be noticed form in Figure 8d,e that there were typical morphologies after reaction. Figure 8f show that the complete dissolution of CFA particles is not achieved even after long reaction times. In addition, the mullite particles with an acicular structure was also found in in sample (S4-4) (Figure 8g,h).

The apparent-shape micrographs and the element composition of the CFA and S4-4 are illustrated in Figure 9. In order to further determine the distribution of each element in the CFA and S4-4, the EDS spectra of two selected areas (labelled 1 and 2) were analysed. From Figure 9a, it can be found that the elemental composition of area 1 included many roughly similar spherical solids, mainly containing the elements of Al, O, Si and Fe (and occasionally K and Ca), which have been reported to reside within the structure of CFA [52]. The spherical solids were cryptocrystalline aluminosilicates which comprised Al, Si and O. As the EDS images suggest (Figure 9i), other elements such as Fe were distributed in a more random pattern. The analysis also shows high concentrations of Na (Figure 9k), but Na is not actually an original element of the CFA. It is thought that calcination using NaOH as an ionised solvent plays an important role in the dissolution of Si^4+^ and Al^3+^ species present in the aluminosilicate matrix of CFA at temperatures above 350 °C [53]. The network of CFA was attacked by sufficient Na^+^ ions in the initial period to further break the Si-O and Al-O bonds. On the other hand, Na^+^ fills the holes in the (SiO_4_) tetrahedron and act as network modifiers, reducing the integrality of the network by replacing bridging oxygens by non-bridging oxygens. In addition, the presence of a high OH^−^ concentration contributes to the release of Si and Al from the CFA [54] and the disruption of inner Si-O and Al-O bonds led to the generation of irregular interconnected units. New Si-Al phases formed after partial disruption of the CFA spheres and re-precipitation at the sphere solution interface. Such newly formed coatings could act as a passivation layer on residues. Hence, it is very difficult for the molten NaOH to act as a liquid media for efficient material transfer during the condensation reaction. The results suggest that NaOH could provide charge compensating ions and facilitate the decomposition of aluminosilicates.

## 4. Conclusions

The main aim of this research was to investigate in-depth the vibrations of the functional groups of CFA during NaOH calcination. The chemical structure of the CFA was found to consist of silicate chains. XRD and FTIR analyses were combined to reveal the decomposition pathways of CFA at the molecular level during the alkali calcination process using NaOH.The aluminosilicate network in the raw CFA gradually disaggregated into clusters with different degrees of reaction during the NaOH calcination process. The number of small clusters increased with temperature, which facilitated structural rearrangements.The results of the FTIR and XRD undoubtedly demonstrated the existence of Si-O-Si linkages in the glass phases of CFA. Pronounced Si-O-Al linkages were present in the structure on account of substitution reactions with a certain portion of the Al in the aluminosilicate. Based on the deconvolution of the main region of interest (800–1300 cm^−1^), several special peaks were identified. The information on the thermal history of the CFA and the chemical and molecular structural changes during the NaOH calcination were clearly identified. The calcination temperature and time were both positively correlated with the reduction of functional groups.In the future, combining current methods with those in other fields, such as alkali calcination followed by hydro-chemical process to synthesize zeolite and alkali calcination followed by recovery of other valuable materials, will contribute to multipurpose use of CFA.

## Figures and Tables

**Figure 1 materials-14-01163-f001:**
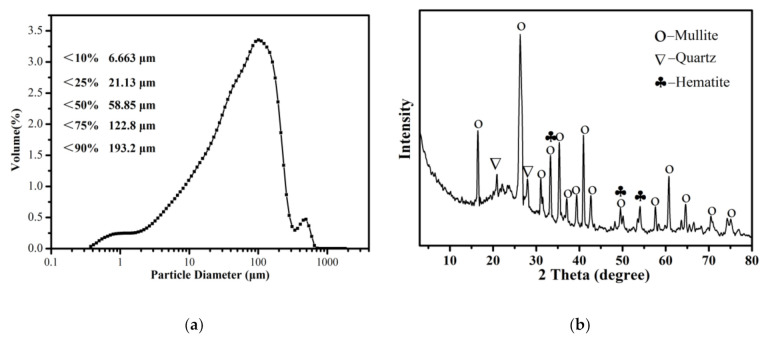
(**a**) The particle size distributions of the raw CFA powder.(**b**) The XRD pattern of the raw CFA powder.

**Figure 2 materials-14-01163-f002:**
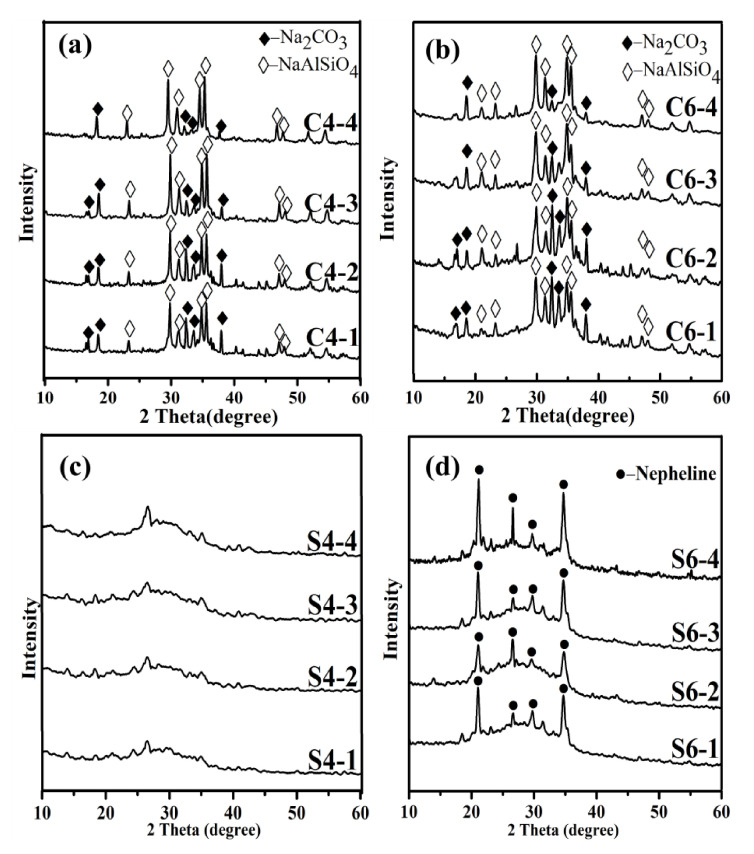
XRD analysis of CFA samples roasted at different temperatures (**a**) 400 °C (**b**) 600 °C and after water leaching of the cake (**c**,**d**).

**Figure 3 materials-14-01163-f003:**
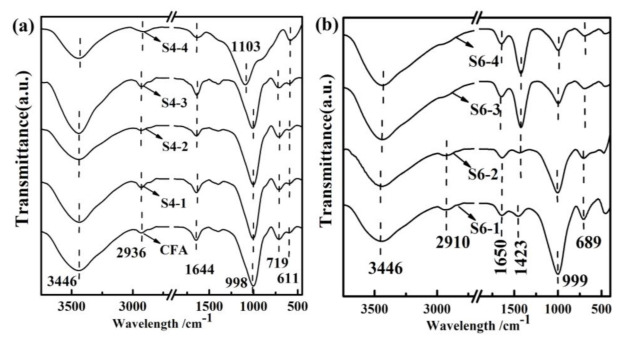
FTIR spectra of (**a**) CFA, and (**b**) alkali calcinated slags after contact with water. CFA was dried at 105 °C for 48 h.

**Figure 4 materials-14-01163-f004:**
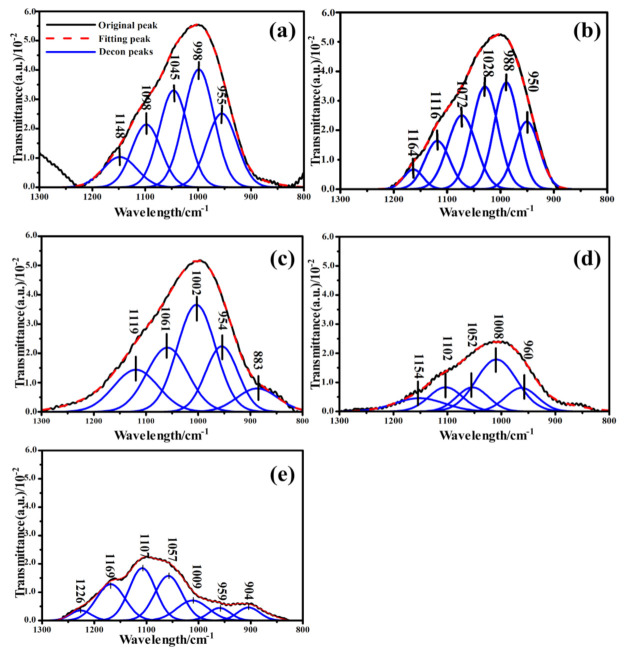
(**a**) Deconvolution of the FTIR spectra of CFA in the region between 1300 and 800 cm^−1^; Deconvolution of the FTIR spectra of samples calcined at 400 °C in the region between 1300 and 800 cm^−1^. The reaction time of (**b**) 0.5 h; (**c**) 1 h; (**d**) 15 h; (**e**) 2 h.

**Figure 5 materials-14-01163-f005:**
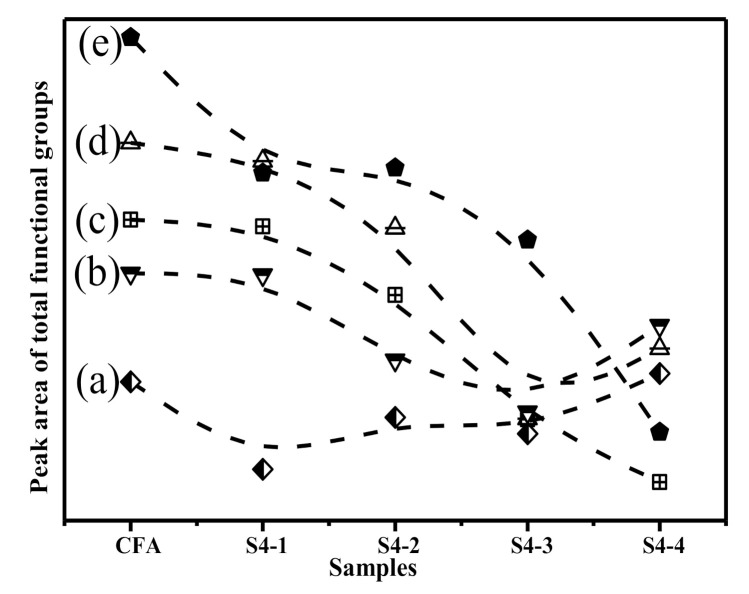
The area under each Gaussian function of samples: (**a**) Asymmetric stretching of (Si,Al)-O-Si in mullite or mullite-like structures. (**b**) Asymmetric stretching of (Si,Al)-O-Si in Q^3^ units; (**c**) Symmetric stretching of Si-O-Si in Q^2^ units; (**d**) Symmetric stretching of Si-O-Si in Q^3^ units; (**e**) Asymmetric stretching of (Si,Al)-O-Si in amorphous glasse.

**Figure 6 materials-14-01163-f006:**
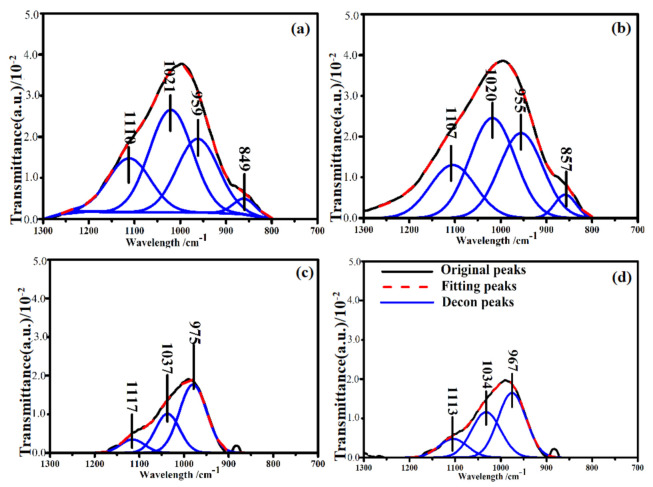
Deconvolution of the FTIR spectra of samples calcined at 600 °C in the region between 1300 and 800 cm^−1^. The reaction time of (**a**) 0.5 h; (**b**) 1 h; (**c**) 1.5 h; (**d**) 2 h.

**Figure 7 materials-14-01163-f007:**
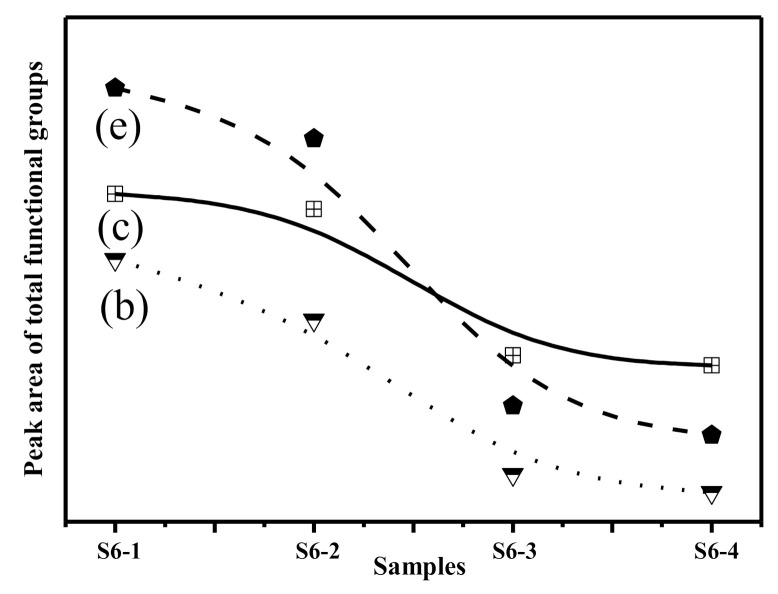
Variation of functional groups after alkaline activation. (**b**) Asymmetric stretching of (Si, Al)-O-Si in Q^3^ units; (**c**) Symmetric stretching of Si-O-Si in Q^2^ units; (**e**) Asymmetric stretching of (Si, Al)-O-Si in amorphous glasses.

**Figure 8 materials-14-01163-f008:**
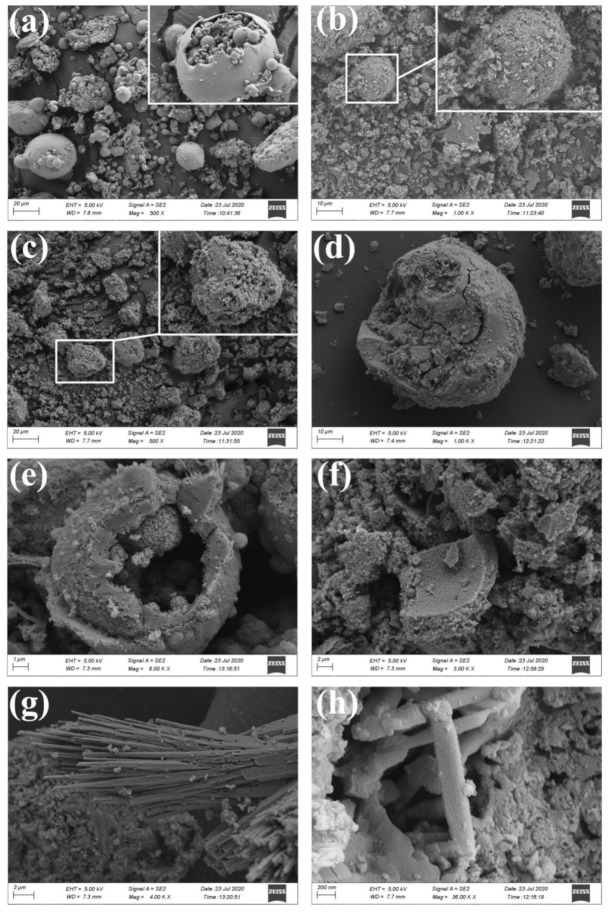
SEM micrographs of the solid samples with variation in calcination time; (**a**) CFA, (**b**) S4-1, (**c**) S4-2, (**d**,**e**), S4-3, (**f**), S4-4. SEM micrographs of the mullite particles with an acicular structure in sample (S4-4) (**g**,**h**).

**Figure 9 materials-14-01163-f009:**
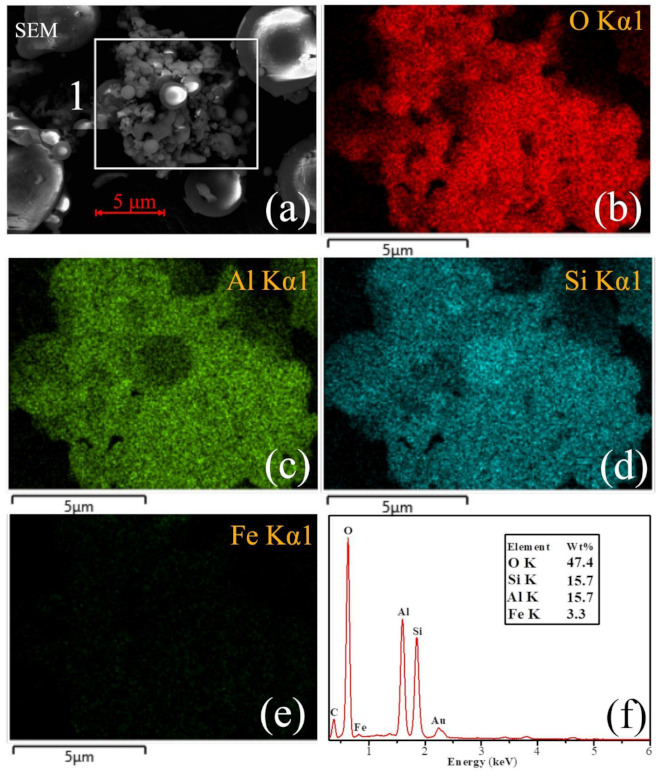
(**a**) SEM images obtained from the raw CFA, (**b**) O distribution map of the area 1, (**c**) Al distribution map of the area 1,(**d**) Si distribution map of the area 1, (**e**) Fe distribution map of the area 1, (**f**) EDS X-ray spectrum from Figure 1 (**a**) area. (**g**) SEM images obtained from the sample S4-4, (**h**) O distribution map of the area 2, (**i**) Al distribution map of the area 2, (**j**) Si distribution map of the area 2, (**k**) Fe distribution map of the area 2, (**l**) EDS X-ray spectrum from Figure 1 (**g**) area.

**Table 1 materials-14-01163-t001:** Chemical composition of CFA.

Component	SiO_2_	Al_2_O_3_	TFe_2_O_3_	MgO	CaO	K_2_O	LOI ^a^
Content (wt.%)	46.23	39.71	3.32	0.18	1.37	0.71	3.41

LOI ^a^: Loss on ignition.

**Table 2 materials-14-01163-t002:** Modification condition of CFA.

Samples Label	Calcination Conditions			Sample Label after Water Leaching
	Mass Ratio CFA:NaOH	Calcination Temperature (°C)	Calcination Time (h)	
**C4-1**	1:4	400	0.5	S4-1
C4-2	1:4	400	1	S4-2
C4-3	1:4	400	1.5	S4-3
C4-4	1:4	400	2	S4-4
C6-1	1:4	600	0.5	S6-1
C6-2	1:4	600	1	S6-1
C6-3	1:4	600	1.5	S6-1
C6-4	1:4	600	2	S6-1

**Table 3 materials-14-01163-t003:** Chemical composition of the sample after water leaching.

Component	SiO_2_	Al_2_O_3_	TFe_2_O_3_	Na_2_O	CaO	K_2_O	MgO
Content (wt.%)	37.61	29.85	3.64	22.37	3.65	1.56	1.32

**Table 4 materials-14-01163-t004:** Assignment of the deconvolved bands [48].

Position/cm^−1^	Assignment	Position/cm^−1^	Assignment
1120–1190	Asymmetric stretching of Si-O-Si [Q^4^]	900–920	Symmetric stretching of Si-O-Si [Q^1^]
1139–1161	^a^ Asymmetric stretching of (Si,Al)-O-Si	~850	Symmetric stretching of Si-O-Si [Q^0^]
1050–1100	Symmetric stretching of Si-O-Si [Q^3^]	795–814	Symmetric stretching of Si-O-Si and stretching of Al-O
1085–1092	^b^ Asymmetric stretching of (Si,Al)-O-Si [Q^3^]	692–730	Symmetric stretching of Al-O in Si (Al)-O- Al linkages
997–1011	^c^ Asymmetric stretching of (Si,Al)-O-Si	612–618	Bending of O-Al-O
1000–950	Symmetric stretching of Si-O-Si [Q^2^]	543–554	Symmetric stretching of Al-O-Si
900–915	Stretching of Si-O-(M ^d^,Me ^e^,Fe)	<461–465	Bending of Si-O-Si and O-Si-O

^a^ in mullite or mullite-like structures, ^b^ in glass (may partially overlap with mullite and quartz); ^c^ in amorphous glasses, could be composed of higher Al concentration, ^d^ M is alkali metal; ^e^ Me is alkali earth metal.

## Data Availability

Data sharing not applicable.

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
