# Peer review of "Study on the Overall Reaction Pathways and Structural Transformations during Decomposition of Coal Fly Ash in the Process of Alkali-Calcination"

_materials, 2021, doi:10.3390/ma14051163_

Round 1
Reviewer 1 Report
The paper “Study on the overall reaction pathways and structural transformations during decomposition of coal fly ash in the process of alkali-calcination” considers phase transformation and the role of NaOH on the structure of coal fly ash during an alkali-calcination process.
I have the following comments:
General comments:
The manuscript deal with an interesting, original and timely topic. The title is self-descriptive and represents the content of the manuscript. The abstract provides a clear view of the content of the paper. The authors properly presented the scientific and technological background of the investigated issues in the introduction section. The conclusions are supported by the obtained results.
Specific comments:
• Provide practical significance of the results obtained in the study. In other words, define some practical suggestions based on the outcome revealed from the study?
• Although the paper is well written and rather easy to read some typos should be removed.
I suggest accepting the paper if the authors succeed in addressing these comments.

Reviewer 2 Report
Manuscript ID: materials-1096475
Title: Study on the overall reaction pathways and structural transformations during decomposition of coal fly ash in the process of alkali-calcination
Authors: Jingjie Yang et al.
The article is devoted to the problem of CFA utilization, which is a very important task, especially in China. The method itself is not new, but the authors used various methods of analysis to clarify the mechanism of roasting CFA with alkali. The article is well-formed and contains many tables and figures. The test results look reliable. However, there are several shortcomings that need to be corrected before final acceptance:
Line 33. This references from 2002. It is not relevant. Chine is the biggest economy in the world. Use new reference.
Line 35. serious environmental issues. Write more specifically what problems of the storage of CFA on the landfills?
Line 78-80. Add references for this research.
Line 95. What kind of coal is used at this power plant? What is the ash content of coal? How much ash is generated per year? How much ash has already been accumulated in the landfill? Did the ash take directly from the filters on a power plant or from the sludge field? Does it coal fly ash of coal bottom ash?
Line 96-97. Authors must add full chemical composition of CFA with LOI, C and S content.
Line 100. The particle size distribution.
Line 112. Did the ash go through the enrichment methods before the experiments (wet magnetic separation or/and froth flotation)?
Table 1. Why use these CFA/NaOH ration and temperatures?
Line 145. Why Authors don’t use TGA/DSC analysis of sample CFA with NaOH addition?
Line 147-169. Add the chemical reactions of roasting and leaching processes.
Figure 2. Re-write the title. This is: XRD analysis of CFA samples roasted at different temperatures (a) 400 °C (b) 600 °C and after water leaching of the cake (c) (d)
Figures 5, 7. Points must be the same colour as curves.
- What is the Al and Si extraction degree after leaching of sintering samples?
- What is the chemical composition of the samples after water leaching?
- How did the particle size distribution, specific surface area, and porosity change after alkaline roasting and leaching? Shoppert et al. show that there is a significant increase in the specific surface area, which will allow using CFA in the future alumina extraction process (https://www.mdpi.com/2075-4701/10/12/1684). After alkali leaching, the mullite particles with an acicular structure was found. Did the authors find similar particles after roasting?
- What is the difference between the article by the authors and Yanxia Guo et al. The role of additives in improved thermal activation of coal fly ash for alumina recovery [doi: 10.1016 / j.fuproc.2012.12.003].
Conclusions should be broken down into several points and contain specific research results with numbers.
References are not in Materials style.
Reviewer 3 Report
Dear authors,
the research paper offers very interesting results from the field of decomposition of coal fly ash in the process of alkali-calcination.
However, I suggest to the authors to make following corrections and additions in the manuscript before its publishing:
1. Abstract part - I suggest state tetrahedral groups instead of "tertrahedron"
- "polymerisation of the CFA" - How can FA polymerise ? Only polymers can...maybe "reaction" will be more suitable expression
- try to also correct or specify in the whole text.
2. Introduction - I do not agree with the claim that most of the CFA produced is disposed of in landfill - it is utilising all over the world in many building sectors
- 83.941m2/g - space is missing between value and unit, and it the rest of paper
- The last sentence of Introduction - "the influence" of what ?
3. Experimental
- CFA - What was the CaO content in FA ?
- It comes from thermal power plant where is the combustion of lignite ?
- Fig. 1 - only Mullite and Quartz were found ? What about hematite ?...axis Volume (%) - space is missing
- Did ypu provide also leaching test of CFA or concentrattion of heavy metals, etc. ? It could improve the paper and better understand some results of FTIR.
- Was this CFA contaminated by SNCR denitrification technology or not ?
- You should in the whole paper put space between value and unit and MPa is with big "Pa"
- line 138 - from 4000 to 400cm-1 - you should write from 400 to 4000...
- line156 - I think that is unlikely so that NaOH could react with CO2 in the air
- line 167 - nepheline has formula (Na,K)AlSiO4. Where does potassium come from?
- line 186 - You stated Si-Ob-Al, but exactly means "b" ?
- Fig. 3 : "Wavelength (cm-1)" - x axis instead of "/"
- Fig. 4 caption is very misleading - (a) CFA - what this exactly means ?
- line 255 - You are writing here about polymerisation of the network polymers of the CFA particles. But CFA particles surely do not content any polymers.
- You should use expression "decomposition" instead of "depolymerisation" - it will be more accurate
- line 290 - "The CFA predominantly had a typical ball-like shape morphology" - it is not true...FA contains in many cases cenospheres
- microspheres within the size range 1–600 µm; Often, You can find less than 1% cenosphere in a fly ash - so it is not a typical morphology of FA!
- Fig. 8 - Why is "Time" with capital letter?
- line 304 - hollow microsphere could not arise durung the reaction of FA with NaOH...
- Fig. 8a - What represent small microsphere particles in the cenosphere ?
Conclusions - You should surely state how the results of the research will be used in practice?
Round 2
Reviewer 2 Report
The authors answered many questions, it is the great job, however, some of them were ignored. I'll repeat them:
- What is the Al and Si extraction degree after leaching of sintering samples?
- What is the chemical composition of the samples after water leaching?
- How did the particle size distribution, specific surface area, and porosity change after alkaline roasting and leaching? Shoppert et al. show that there is a significant increase in the specific surface area, which will allow using CFA in the future alumina extraction process (https://www.mdpi.com/2075-4701/10/12/1684). After alkali leaching, the mullite particles with an acicular structure was found. Did the authors find similar particles after roasting?
It seems to me that in this article it is necessary to add links to study of hydrometallurgical alkaline ash silica removal as an example of an alternative to roasting CFA by alkali:
https://doi.org/10.1016/j.ultsonch.2019.104765
https://doi.org/10.3390/met10121684
https://doi.org/10.1021/acsomega.0c04737
I will leave my recommendation (major revision). However, if the authors answer my questions I am immediately ready to accept the article.
Round 3
Reviewer 2 Report
The authors answered all questions. However, the style of the attached article file does not correspond to the Materials style. There must have been an error and the Authors have attached the wrong file.
Why didn't the authors add the answers to the questions in the article? Data on the aluminum and silicon extraction degree, as well as the chemical composition CFA after leaching (Mg2O must change to MgO) and a SEM-image of acicular mullite - all of this is very valuable information and will significantly improve the quality of the article.
I ask the Authors to add their answers to the article (including a SEM-image of the acicular mullite).
